# Quinagolide Treatment Reduces Invasive and Angiogenic Properties of Endometrial Mesenchymal Stromal Cells

**DOI:** 10.3390/ijms23031775

**Published:** 2022-02-04

**Authors:** Corinne Iampietro, Alessia Brossa, Stefano Canosa, Stefania Tritta, Glenn E. Croston, Torsten Michael Reinheimer, Filippo Bonelli, Andrea Roberto Carosso, Gianluca Gennarelli, Stefano Cosma, Chiara Benedetto, Alberto Revelli, Benedetta Bussolati

**Affiliations:** 1Department of Molecular Biotechnology and Health Sciences, Università degli Studi di Torino, 10126 Torino, Italy; corinneiampi@hotmail.it (C.I.); alessia.brossa@unito.it (A.B.); stefania.tritta@unito.it (S.T.); bonelli.filippo@hsr.it (F.B.); 2Obstetrics and Gynecology 1U, Physiopathology of Reproduction and IVF Unit, S. Anna Hospital, Department of Surgical Sciences, Università degli Studi di Torino, 10126 Torino, Italy; stefano.canosa@unito.it (S.C.); andrea88.carosso@gmail.com (A.R.C.); gennarelligl@gmail.com (G.G.); stefano.cosma@unito.it (S.C.); chiara.benedetto@unito.it (C.B.); alberto.revelli@unito.it (A.R.); 3EicOsis, Davis, CA 95616, USA; glenncrostonphd@gmail.com; 4Reinheimer.Expert ApS, 2300 Copenhagen, Denmark; tmr@reinheimer.expert

**Keywords:** mesenchymal stromal cells, endometriosis, quinagolide, dopamine receptor agonist, endothelial differentiation, invasion

## Abstract

Endometrial mesenchymal stromal cells (E-MSCs) extensively contribute to the establishment and progression of endometrial ectopic lesions through formation of the stromal vascular tissue, and support to its growth and vascularization. As E-MSCs lack oestrogen receptors, endometriosis eradication cannot be achieved by hormone-based pharmacological approaches. Quinagolide is a non-ergot-derived dopamine receptor 2 agonist reported to display therapeutic effects in in vivo models of endometriosis. In the present study, we isolated E-MSCs from eutopic endometrial tissue and from ovarian and peritoneal endometriotic lesions, and we tested the effect of quinagolide on their proliferation and matrix invasion ability. Moreover, the effect of quinagolide on E-MSC endothelial differentiation was assessed in an endothelial co-culture model of angiogenesis. E-MSC lines expressed dopamine receptor 2, with higher expression in ectopic than eutopic ones. Quinagolide inhibited the invasive properties of E-MSCs, but not their proliferation, and limited their endothelial differentiation. The abrogation of the observed effects by spiperone, a dopamine receptor antagonist, confirmed specific dopamine receptor activation. At variance, no involvement of VEGFR2 inhibition was observed. Moreover, dopamine receptor 2 activation led to downregulation of AKT and its phosphorylation. Of interest, several effects were more prominent on ectopic E-MSCs with respect to eutopic lines. Together with the reported effects on endometrial and endothelial cells, the observed inhibition of E-MSCs may increase the rationale for quinagolide in endometriosis treatment.

## 1. Introduction

Endometriosis is a reproductive age-associated disease characterized by the presence of endometrial-like tissue outside the uterus [1,2]. Sex-steroid hormones are not only key players for the maintenance of normal uterine function and fertility, but they also regulate the growth of ectopic lesions causing periodic bleeding and inflammation associated with pelvic pain and infertility [3]. Importantly, stem cell activity in the basalis of the endometrium plays a critical role in endometrial function, supporting cyclic regeneration after menstruation [4,5,6]. In particular, local endometrial mesenchymal stromal cells (E-MSCs) have been isolated and characterized in several works [7], stimulating interest in their potential in endometrium regeneration. E-MSCs recapitulate the majority of mesenchymal stem cell properties [8], including clonogenicity, multipotency and a specific surface phenotype that distinguish them from leukocytes, hematopoietic and endothelial cells [9,10]. Moreover, E-MSCs show strong self-renewal in vitro [10] and capacity to regenerate endometrial stromal vascular tissue in vivo [5,11], as well as strong motility feature and invasiveness [12]. In addition, E-MSCs represent a heterogenic population of mesenchymal stem cells and stromal fibroblasts, sharing a number of markers and functions. The specific expression of CD146, platelet-derived growth factor-receptor beta (PDGFRB) and sushi domain containing-2 (SUSD2) revealed E-MSCs pericyte identity and perivascular localization, respectively [13,14].

According to the guidelines provided by the European Society of Human Reproduction and Embryology, medical treatment for endometriosis-associated pain and infertility is based on the administration of anti-inflammatory drugs coupled with agents acting on the hormonal alteration of the menstrual cycle to produce chronic anovulation and an overall hypoestrogenic environment [15]. However, as E-MSCs lack oestrogen receptors, endometriosis eradication cannot be achieved by hormone-based pharmacological approaches. Anti-angiogenic drugs are currently of increasing interest in consideration of the role of vasculogenesis and angiogenesis in the progression and maintenance of endometriotic lesions [16].

Indeed, the pro-angiogenic environment has a critical role in the implantation, maintenance and growth of endometriotic implants, as supported by the significant increase in vascular endothelial growth factor (VEGF) levels in the peritoneal fluid of women with endometriosis and in ectopic endometrial tissue [17,18,19,20]. We previously isolated E-MSCs from endometriotic lesions and showed their increased pro-angiogenic properties, including VEGF release, with respect to eutopic E-MSCs and the related inhibitory effect of a tyrosine kinase inhibitor, Sorafenib [21]. In this context, dopamine and its receptor agonists may represent an alternative to current anti-angiogenic agents due to the inhibition of VEGF release and VEGF receptor 2 (VEGFR-2) activation [22]. Among the non-ergot dopamine receptor 2 (DRD2) agonists, quinagolide has been successfully tested in an experimentally induced endometriosis rat model [23], and quinagolide tablets (Norprolac^R^) have been marketed for treatment of hyperprolactinemia for a long time with substantial clinical experience and safety data [24]. Moreover, quinagolide is currently undergoing two different phase 2 trials investigating the effect of drug-releasing vaginal rings in women with endometriosis (NCT03749109, NCT03692403). Recently, it was shown that the DRD2 agonist cabergoline reduced the angiogenic potential of E-MSCs in an endothelial co-culture setting [25]. However, the functional effects of the dopamine receptor agonist quinagolide on eutopic and ectopic E-MSCs have not been evaluated, and neither has the relative involvement of DRD2 and VEGFR2 pathway modulation. In the present study, we evaluated the effects of quinagolide treatment on relevant functional characteristics of eutopic and ectopic E-MSC lines, isolated from normal endometrium or endometriotic lesions, including proliferation, invasion and endothelial differentiation, and the related molecular mechanisms involved.

## 2. Results

### 2.1. Generation and Characterization of Eutopic and Ectopic E-MSC Lines

A cohort of ten patients was enrolled for the study, including control (*n* = 3), ovarian (*n* = 6) and peritoneal (*n* = 3) endometriosis. The demographic and clinical aspects of the patient population are summarized in Table 1.

In particular, stromal cells from eutopic and ectopic tissues were isolated, as reported in Materials and Methods, and cultured in EBM. After seven days, culture medium was refreshed, allowing the removal of dead and/or unselected cells and promoting the clonal growth of E-MSCs. Generated cell lines were analysed for their fibroblastic phenotype, adherence to plastic, and surface marker expression (Figure 1A,B). FACS analysis showed the mesenchymal phenotype of all E-MSC lines. In particular, expression of mesenchymal markers CD44, CD73, CD90 and CD29 was similar in eutopic and ectopic E-MSCs, with only a statistically significant increase of CD105 in both ovarian and peritoneal ectopic E-MSCs compared to eutopic ones. Moreover, cell contamination was excluded by the lack of the epithelial marker EPCAM and the endothelial/hemopoietic markers CD34, CD45 and Tie2. All E-MSC lines were positive for specific endometriotic mesenchymal stem cell markers SUSD2 and PDGFRb (CD140b), with lower expression by ectopic E-MSCs with respect to eutopic ones, suggesting that E-MSCs represent a heterogenic population of mesenchymal stem cells and stromal fibroblasts, sharing a number of markers and functions. In selected experiments, E-MSC lines were SUSD2 sorted to possibly enrich for the E-MSCs with respect to the stromal cells. By FACS analysis, the resulting SUSD2^+^ E-MSC lines showed the same phenotypic profile compared to the original E-MSCs (not shown) and SUSD2 expression returned to basal level after 1 culture passage.

### 2.2. Quinagolide Effect on the Invasion Potential of E-MSCs

E-MSC lines were used to evaluate the effect of quinagolide, a dopamine receptor agonist, on E-MSC functional properties. As dopamine receptor agonists can inhibit VEGF-induced VEGFR-2 activity [22,26], we first evaluated the expression of both the quinagolide receptor DRD2 and of VEGFR-2 on E-MSCs (Figure 2). Quinagolide receptor DRD2 was expressed by all E-MSC lines regardless of the passage number or the SUSD2 enrichment (Figure 2A). Moreover, at the mRNA level, DRD2 expression was significantly increased in ectopic lines (Figure 2B). Quinagolide treatment reduced DRD2 mRNA expression in ectopic lines, suggesting an effect on receptor downregulation (Figure 2C). No expression of VEGFR2 was observed in E-MSCs (not shown), as previously reported [25].

A concentration response curve showed a lack of quinagolide effect on E-MSC proliferation and apoptosis (Figure 3). As shown in Figure 3A, different concentrations of quinagolide did not increase apoptosis of E-MSCs after 24, 48 and 72 h. Moreover, quinagolide treatment did not affect proliferation of E-MSCs after 24 h (Figure 3A). Similarly, HUVEC cells, used as a control, did not show alteration in apoptosis and proliferation after 24 h of quinagolide treatment (Figure 3B). Based on these results, 100 nM of quinagolide was chosen for further experiments.

We subsequently evaluated the effect of quinagolide on E-MSC migration and invasiveness using an invasion assay. As shown in Figure 4A, eutopic and ectopic E-MSCs were able to invade Matrigel after 48 h in culture. Interestingly, invasion of both eutopic and ectopic E-MSC lines was significantly reduced after 48 h quinagolide treatment (Figure 4B), in a dose-dependent manner (Figure 4C). Moreover, the quinagolide effect was completely reverted by pre-treatment with the DRD2 receptor antagonist spiperone (Figure 4D), indicating that the observed anti-invasive effect of quinagolide was DRD2 dependent.

### 2.3. Quinagolide effect on the Endothelial Differentiation of E-MSCs

Considering the reported ability of E-MSCs to differentiate into endothelial cells, we tested the effect of quinagolide in an E-MSC-HUVEC co-culture model, testing its in vitro endometriosis angiogenic potential. Using this model, after 48 h of direct co-culture of E-MSCs and HUVECs, CD31 expression was acquired by E-MSCs, confirming the influence of HUVECs in the differentiation potential of E-MSCs into endothelial cells (Figure 5). The use of HUVECs traced by GFP expression (>98% in all experiments) allowed us to easily separate, by a selective FACS gating strategy, the CD31^+^/GFP^+^ HUVECs and the high CD31^+^/GFP^−^ E-MSC population that results after co-culture (Figure 5A). After 24 h of direct co-culture assembly, cells were then treated with quinagolide and incubated for 24 h before analysis. As cabergoline treatment (25 µM) was previously described to decrease the E-MSC’s angiogenic potential, this drug was used as positive control [25] (Figure 5B). The typical increase in the percentage of CD31 expressing E-MSCs evaluated after co-culture was significantly reduced by a 24 h quinagolide treatment in ectopic E-MSCs, while no significant alteration was measured in eutopic ones (Figure 5C).

To confirm that the effect observed was due to the quinagolide treatment through its DRD2 receptor, direct co-cultures were treated with the specific DRD2 receptor antagonist spiperone one hour before quinagolide treatment. As shown in Figure 5D, the quinagolide effect was blocked by spiperone pre-treatment with no reduction in the percentage of CD31 expressing E-MSCs, confirming that the effect observed on E-MSCs was DRD2-mediated. Sorafenib and cabozantinib, anti-angiogenic tyrosine kinase inhibitors, had no effect on endothelial differentiation (Figure 5D), excluding the involvement of VEGFR-2 in this process. Moreover, quinagolide did not reduce the levels of VEGF in the co-culture model (Appendix A).

### 2.4. Molecular Mechanisms Related to Quinagolide Effect

In order to explain the molecular mechanisms at the basis of quinagolide’s effect on E-MSCs, the AKT pathway was analysed after quinagolide treatment (Figure 6). Quinagolide treatment reduced total AKT levels in ectopic E-MSCs (Figure 6A). We observed a significant decrease in AKT phosphorylation in both eutopic and ectopic cell lines treated with quinagolide for 24 h (Figure 6B). Moreover, quinagolide reduced the AKT phosphorylation when added to E-MSC-HUVEC co-cultures (Figure 6C). These data suggest the effect of DRD2 agonists on AKT signalling. The effect on both AKT levels and on phosphorylation were more evident on ectopic E-MSC lines, with respect to eutopic ones, possibly in accordance with the increased expression of DRD2 on these lines.

## 3. Discussion

E-MSCs are postulated to play a critical role in the pathogenesis of endometriosis, contributing to the establishment and progression of ectopic lesions and supporting the vascularization and growth of the endometrial stromal tissue [27]. In the present study, we demonstrated that quinagolide inhibited the invasive properties of E-MSCs, and limited their endothelial differentiation in an endothelial co-culture model of angiogenesis.

Quinagolide is a non-ergot-derived DRD2 agonist [28], described to be a safe and well-tolerated drug in long-term prolactinoma treatment, without severe side effects and with advantages when compared to other dopamine agonists [24,29]. Comparison of the dopaminergic D2 receptor binding properties of different agonists (quinagolide, bromocriptine, pergolide and cabergoline) indicated quinagolide as the most potent DRD2 agonist, with EC50 at 0.058 nM [30]. The first pilot study evaluating the possible use of quinagolide for endometriosis treatment involved patients simultaneously suffering from severe endometriosis and hyperprolactinemia [31]. Quinagolide treatment reduced the size of endometriotic lesions, possibly by acting through VEGFR-2 downregulation [31]. At present, the effect of quinagolide in endometriosis is under investigation in phase 2 clinical trials (NCT03749109, NCT03692403). Moreover, quinagolide has been already shown to reduce endometriotic lesions in rodents [32], and to induce a significant regression in endometriotic implants and reduced levels of IL-6 and VEGF in peritoneal fluid [23]. Similarly, quinagolide reduced the angiogenesis in a mouse model of endometriosis, reducing the size of active endometriotic lesions, cellular proliferation and VEGF levels [33]. The anti-angiogenic effects were comparable to those of anti-VEGF therapy [34]. These reported effects may result from a combined activity of quinagolide on endothelial and endometrial cells, through signalling pathways linked to DRD2 activation and, in endothelial cells, to VEGFR-2 inhibition through dephosphorylation [22,26]. The mRNA and protein expression of DRD2 and VEGFR-2 was previously observed in both eutopic and ectopic fragments implanted in nude mice [35].

In this study, aiming to evaluate a pivotal effect of quinagolide on E-MSCs, we confirmed DRD2 expression in E-MSCs isolated from both eutopic normal peritoneal tissue and ectopic (peritoneal and ovarian) endometrial lesions. Interestingly, DRD2 levels appeared to be slightly higher in endometriotic E-MSCs. On the cell surface, DRD2 may co-localize with VEGFR-2 [22], and its activation may consequently limit VEGFR-2 phosphorylation and promote its endocytosis in endothelial cells. However, the lack of VEGFR-2 we observed on E-MSCs may suggest that quinagolide’s effect does not involve VEGFR-2. It did not show any impact on proliferation and apoptosis, whereas a dose-dependent activity of quinagolide was observed on invasion inhibition, suggesting a possible therapeutic use in the reduction of endometriosis spread outside the uterine cavity. These results are consistent with the previously reported inhibitory effects of dopamine agonists on cancer cells and skin mesenchymal stem cell migration [36,37].

In addition, quinagolide was able to inhibit E-MSC endothelial differentiation. We previously reported a model of E-MSC differentiation with activation of a number of endothelial genes when co-cultured with endothelial cells [25]. Herein, we observed that quinagolide was able to reduce E-MSC differentiation, evaluated as the acquisition of the endothelial marker CD31. Importantly, quinagolide’s effect was more prominent on ectopic rather than eutopic E-MSCs when added to the co-culture. This could be related to the increased expression of DRD2 on ectopic E-MSCs.

The inhibitory effect of spiperone, a selective DRD2 antagonist, confirmed that the observed anti-invasive and anti-angiogenic effects of quinagolide were dependent on DRD2 activation. Moreover, the observed effect was independent from the inhibition of VEGF release. Indeed, this model was independent of soluble factor release, and was previously shown to require cell contact [25]. Accordingly, quinagolide did not affect VEGF release. Moreover, sunitinib and cabozantinib, tyrosine kinase inhibitors blocking activation and signalling of growth factor receptors [38], including VEGFR-2, did not affect E-MSC endothelial differentiation, further supporting the role of DRD2 in E-MSC endothelial differentiation as well.

Focusing on putative VEGFR-2 independent signalling pathways downstream of dopamine receptors, we evaluated AKT activity, previously reported as modulated by direct receptor activation [39]. Previous studies have convincingly shown that the AKT pathway mediates dopaminergic activities, and that manipulations of the AKT/GSK3 pathway results in significant alterations in dopamine-related functions and behaviors [40]. In the brain in particular, activation of DRD2 may lead to a beta-arrestin mediated deactivation of AKT [40] and decrease its phosphorylation, leading to a reduction of AKT activity [41]. A specific DRD2 activation was also able to reduce the migration of skin MSCs to the wound beds by suppressing AKT phosphorylation [36]. We also found that quinagolide treatment of E-MSCs or of E-MSC/HUVEC co-cultures decreased AKT phosphorylation. Moreover, beside phosphorylation, AKT protein levels were reduced. Interestingly enough, ectopic E-MSC lines showed a better response to quinagolide in terms of AKT downregulation and deactivation, in accordance with the differential presence of DRD2 receptors and with the functional effect on the different E-MSC lines. These results confirmed the differential proliferation, migration, and angiogenic ability of ectopic E-MSCs reported with respect to eutopic E-MSCs from the same patient or from healthy patients [20,21]. The different DRD2 expression and behavior of E-MSCs might be due to selection and/or epigenetic modulation of the extrauterine microenvironment found in ectopic sites, as reported for cancer lesions [42].

A limitation of the study was the impossibility to obtain a pure SUSD2+ mesenchymal stem population, as the sorted SUSD2+ cells spontaneously lost the marker expression after culture. Indeed, this confirms the previously reported observation that cultured E-MSCs spontaneously differentiate into fibroblasts, and that E-MSC and fibroblast populations represent a continuum, and share characteristics and several functions [25].

## 4. Materials and Methods

### 4.1. Patients

A total of 10 patients were enrolled in the present study for tissue collection and subsequent cell isolation between September 2018 and January 2020. All patients provided preoperative written informed consent before receiving endometrial sampling or surgery for treatment of ovarian or/and peritoneal endometriosis in the Department of Surgical Sciences at the University of Turin, after approval by the Ethics Review Board of the Health and Science City of Torino (Città della Salute e della Scienza di Torino. N°0055438, 28 May 2018). Inclusion criteria for control patients were: age ≤ 42 years, normal body mass index (BMI 18–25), regular menses, absence of uterine pathologies. Inclusion criteria for patients with endometriosis were: painful stage III or IV endometriotic lesion (as classified by the American Society for Reproductive Medicine), no response to hormonal therapy treatment for at least 6 months, presence of ovarian endometrioma with diameter > 4 cm.

### 4.2. Endometriotic Specimen Collection and E-MSC Isolation

Three eutopic samples were collected by gently scraping the endometrium of control patients, used as controls, whereas the other nine ectopic samples were obtained by surgical biopsy of the inner wall of the ovarian or peritoneal endometrial tissue of endometriotic patients. In two patients, both ovarian and peritoneal endometrial samples were collected since the patients presented the two different types of endometriosis. The tissues (around 0.5 cm^3^) were immediately processed by dissection into small fragments in a sterile tissue culture dish using a sterile scalped blade in a laminar flow hood, as previously described [21,25]. Briefly, the fragments were first enzymatically digested with 0.1% Type I Collagenase (Sigma-Aldrich, St. Louis, MO, USA) for 30 min in a 37 °C heater, and then they were mechanically disaggregated through 60 mm and 120 mm meshes. After two 10 min centrifugations at 1500× *g* for washing, the pellets were resuspended in EBM plus supplement kit (Lonza) as described for E-MSC isolation [21], and cells were seeded in T25 flasks. Dead cells were poured off 72 h later and cell clones were typically observed after 5–7 days, but medium was changed only after 7 days to guarantee cell attachment. Then, medium was recovered every 2–3 days and cells were passaged for the first time 10–14 days after plating, when confluence was reached. In the subsequent passages, cells were split two times per week. Twelve E-MSC lines were isolated (eutopic E-MSCs *n* = 3, ectopic ovarian E-MSCs *n* = 6, ectopic peritoneal E-MSCs *n* = 3) and cultured for a maximum of 11 passages to evaluate their proliferative ability. All the experiments were performed between passages 1–7.

### 4.3. Flow Cytometric Analysis

E-MSCs were characterized at passage 1 or 2 using FACS Celesta (BD Biosciences, Frankin Lakes, NJ, USA). Cells, detached using a non- enzymatic cell dissociation solution (Sigma-Aldrich, St. Louis, MO, USA), were resuspended in 100 µL of 0.1% BSA-PBS (Sigma-Aldrich). For each staining, 100,000 cells were incubated for 20 min at 4 °C with FITC, APC or PE-conjugated antibodies against: CD29, CD44, CD73, CD90 (BD Biosciences); CD31, CD34, CD105, CD140b, CD146, TEK receptor tyrosine kinase (Tie2), Sushi domain-containing protein 2 (SUSD2) (Miltenyi Biotech, Bergisch Gladbach, Germany); CD45 (AbD Serotec, Bio-Rad, Hercules, CA, USA); or epithelial cell adhesion molecule (EPCAM) (BioLegend, San Diego, CA, USA). Labelled cells were washed by centrifugation and final pellet was resuspended in 200 µL of 0.1% BSA-PBS. Isotype (Miltenyi Biotec) was used as negative control.

### 4.4. Protein Extraction and Western Blot

Cell pellets were lysed and western blot performed as described [21]. The following primary antibodies were used: anti-DRD2, anti-Vinculin (Sigma-Aldrich), anti-AKT and anti-P-AKT (both from Cell Signalling, Danvers, MA, USA). After rinsing in wash buffer (0.1% Tween in PBS) horseradish peroxidase-conjugated secondary antibodies (Thermo Scientific, Waltham, MA, USA) were used for 1 h at 1:3000 dilutions. Membranes were finally washed and incubated with ECL chemiluminescence reagent (Bio-Rad, Hercules, CA, USA) in a Chemidoc machine (Bio-Rad).

### 4.5. RNA Isolation and Real-Time PCR

Trizol Reagent (Ambion, Thermo Scientific, Waltham, MA, USA) was used to isolate total RNA of different cell preparations, according to the manufacturer’s protocol. RNA was then quantified spectrophotometrically using Nanodrop ND-1000. Quantitative real-time PCR was performed for gene expression analysis. Briefly, using the HighCapacity cDNA Reverse Transcription Kit (Applied Biosystems, Waltham, MA, USA), first-strand cDNA was produced from 200 ng of total RNA. Real-time PCR experiments were then performed in a 20 µL reaction mixture containing 5 ng of cDNA template, the sequence-specific oligonucleotide primers (all purchased from MWG-Biotech, Eurofins Genomics, Ebersberg, Germany) and the Power SYBR Green PCR Master Mix (Applied Biosystems). GAPDH mRNA was used to normalize RNA inputs. Fold change expression with respect to control was calculated for all samples.

### 4.6. Drugs and Reagents

Quinagolide powder (provided by Ferring Pharmaceuticals, Saint-Prex, Switzerland) was stored at 4 °C and resuspended in dimethylsuphoxide (DMSO) to a stock solution of 1 mM immediately before use. Spiperone powder (Sigma-Aldrich, St. Louis, MO, USA) was resuspended in water to a stock concentration of 1 mM. Cabergoline powder and Sorafenib and Cabozantinib (Sigma-Aldrich) were resuspended in DMSO to a final concentration of 25 mM or 10 mM, respectively, according to the manufacturer’s instructions. Quinagolide, spiperone and cabergoline were diluted to a final concentration of 100 nM, 5 μM and 25 μM, respectively. Quinagolide, cabergoline, sorafenib and cabozantinib were administered for 24 h during co-culture experiments, while spiperone was administered 1 h before quinagolide treatment. For invasion assays, E-MSCs were treated with spiperone 1 h before the cell detachment and quinagolide was added to Matrigel plated cells.

### 4.7. Cell Proliferation Assay

Cells were plated in growth medium at a concentration of 2500 HUVECs/well and 3000 E-MSCs/well in a 96-multiwell plate. The day after, quinagolide was added to the growth medium at different concentrations after 24 h. Deoxyribonucleic acid synthesis was detected as incorporation of 5-bromo-2-deoxyuridine (BrdU) into the cellular DNA after 48 h from cell plaiting, using an enzyme-linked assay kit (Sigma-Aldrich). Untreated cells were used as control. Data are expressed as the mean ± standard deviation (SD) of the media of absorbance of at least three different experiments and normalized to control.

### 4.8. Apoptosis

Annexin V assays were performed using the Muse^TM^ Annexin V & Dead Cell Kit (Luminex corporation, Austin, TX, USA) according to the manufacturer’s recommendations. Briefly, 20 × 10^3^ cells were plated and, after 24 h, treated with different concentrations of quinagolide. After 24, 48 and 72 h, cells were detached and resuspended in Muse^TM^ Annexin V & Dead Cell Kit and the percentage of apoptotic cells (Annexin V^+^) was measured. Data are expressed as the mean ± standard deviation (SD) of the media of absorbance of at least three different experiments and normalized to control.

### 4.9. Invasion Assay

E-MSCs were seeded in triplicate in Matrigel-precoated (100 μg Matrigel/transwell) 8 µm pore transwells at a concentration of 50,000 cells per well in 200 µL of RPMI 2% FCS with/without quinagolide at the indicated concentration. To test the DRD2 antagonist effect, E-MSCs were pre-treated with spiperone (5 µM) for 1 h at 37 °C before cell detachment. After 48 h, invaded E-MSCs on the bottom side of the transwell were fixed with methanol and stained with crystal violet. At least five pictures per transwell were acquired (original magnification: 100×), and the percentage of transwell area covered by invaded E-MSCs was quantitatively measured by ImageJ software (ImageJ, U.S., National Institutes of Health, Bethesda, MA, USA).

### 4.10. HUVEC-E-MSCs Endothelial Differentiation in Co-Culture

HUVECs derived from the umbilical vein vascular wall were plated on fibronectin-coated flasks and grown in endothelial cell basal medium with an EGM-MV kit (Lonza, Basilea, Switzerland; containing epidermal growth factor, hydrocortisone, bovine brain extract) and 10% fetal calf serum in 37 °C and 5% CO_2_ atmosphere incubator. Cells were transduced with lentiviral particles containing pGIPZ lentiviral vector (Open Biosystems, Thermo Scientific, Waltham, MA, USA) expressing green fluorescent protein (GFP). In particular, 293T cells were first transfected with pGIPZ construct adopting the ViralPower Packaging Mix (Life Technologies, Carlsbad, CA, USA) and then the lentiviral stock was titered. HUVEC transduction was performed at the first passages and at 70% cell confluence following the manufacturer’s instructions. After Puromycin (ThermoFisher) (1000 ng/mL) selection, antibiotic-resistant HUVECs were expanded. Finally, FACS analysis was performed to evaluate the expression of endothelial markers and GFP + >98% (Appendix A).

E-MSC endothelial differentiation assay was performed as previously described [25]. Briefly, an indirect co-culture assembly was obtained by plaiting HUVECs and E-MSCs at a ratio of 1:1 (1.5 × 10^4^/cell line) in E-MSC medium in T25 and maintaining the co-culture for 48 h in 37 °C and 5% CO_2_ atmosphere incubator. HUVECs and E-MSCs cultured alone were used as control for each experiment. E-MSCs were gated as GFP negative population, whereas HUVECs were gated as GFP positive population. CD31 expression was evaluated on the described gated cells as percentage of CD31 APC positive events.

### 4.11. Statistics

The number of patients enrolled in this study was set as three for condition. Three cell lines were generated from control eutopic tissue and peritoneal endometriotic lesion and 6 from ovarian peritoneal endometriotic lesions, and each experiment was performed at least in triplicate. Data are shown as mean ± SD. Two-tail Student’s *t* test was used for analysis when two groups of data were compared, while 2-way ANOVA with Dunnett’s multiple comparison test was applied when comparing more than two groups of data. All statistical analyses were done with GraphPad Prism software version 6.0 (GraphPad Software, Inc., San Diego, CA, USA). *p*-Values of < 0 .05 were considered significant.

## 5. Conclusions

In conclusion, we reported the effect of a DRD2 agonist, quinagolide, on E-MSC lines for the first time, showing its effect on reduction of invasion and endothelial differentiation trough the AKT signalling pathway. Together with the reported effects on endometrial and endothelial cells, the observed prominent inhibitory effect of quinagolide on E-MSC ectopic cell lines further supports the rationale for use of this drug in endometriosis treatment.

## Figures and Tables

**Figure 1 ijms-23-01775-f001:**
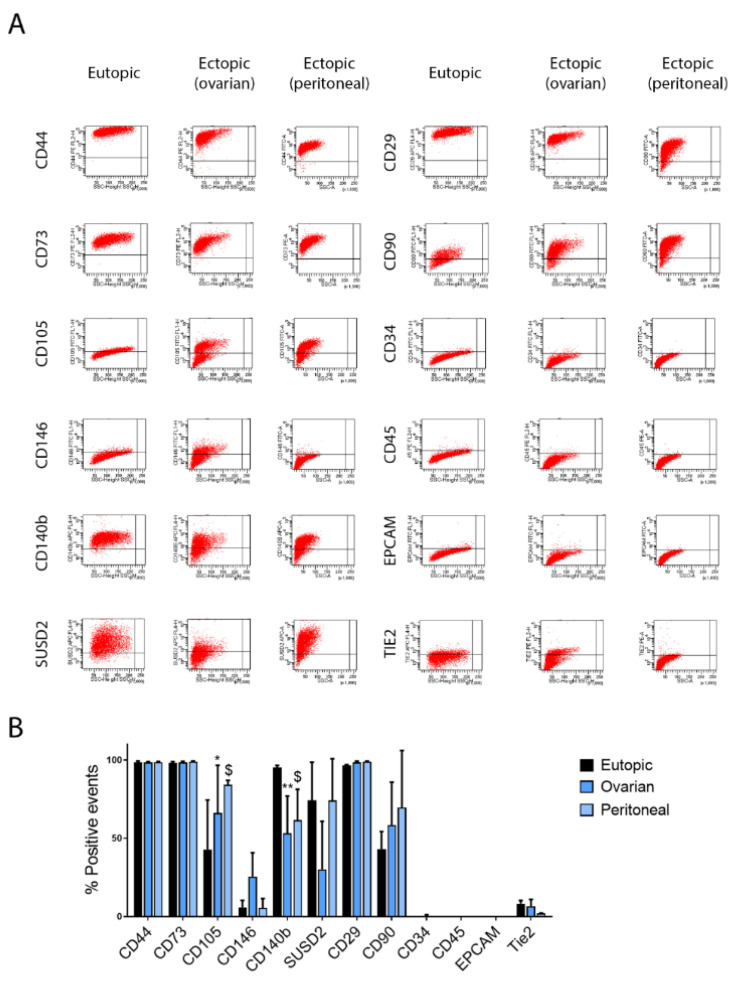
Expression of mesenchymal, hematopoietic, endometriotic, epithelial and endothelial markers by human eutopic and ectopic E-MSCs. Representative FACS analysis (**A**) and quantification (**B**) of eutopic and ectopic (both ovarian and peritoneal) E-MSCs lines for the expression of mesenchymal (CD44, CD73, CD90, CD29, CD105, CD146), hematopoietic (CD45, CD34), endometriotic (SUSD2, CD140b), epithelial (EPCAM) and endothelial (TIE2) markers. Analyses were performed on every cell line used in the study between passage 1 and 2. Data are shown as mean ± SD of all the tested lines: eutopic (*n* = 3), ovarian (*n* = 6) and peritoneal (*n* = 3) ectopic E-MSCs. P: *p*-value; * = *p* < 0.05, ** = *p* < 0.01 (ovarian vs. eutopic); $ = *p* < 0.05 (peritoneal vs. eutopic).

**Figure 2 ijms-23-01775-f002:**
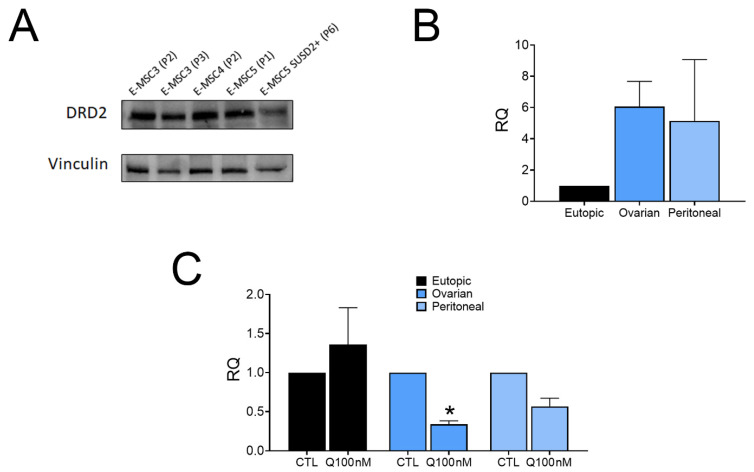
Effect of quinagolide on DRD2 expression in eutopic and ectopic E-MSCs. (**A**) Western blot analysis showing the presence of DRD2 in E-MSC lines, at different passages, and in sorted SUSD2+ E-MSCs. (**B**) Real-Time PCR analysis showing the relative quantification (RQ) of DRD2 mRNA expression by eutopic and ectopic E-MSCs. Data are represented as mean ± SD of three different eutopic or ectopic (ovarian and peritoneal) lines, and normalized to GAPDH and to eutopic E-MSCs. (**C**) Real-Time PCR analysis showing DRD2 mRNA expression after 48 h of 100 nM quinagolide treatment (Q100 nM) by eutopic and ectopic E-MSCs. Data are represented as mean ± SD of three different eutopic or ectopic (ovarian and peritoneal) lines, and normalized to GAPDH and to untreated E-MSCs (CTL). ANOVA was performed: * = *p* < 0.05 vs. CTL.

**Figure 3 ijms-23-01775-f003:**
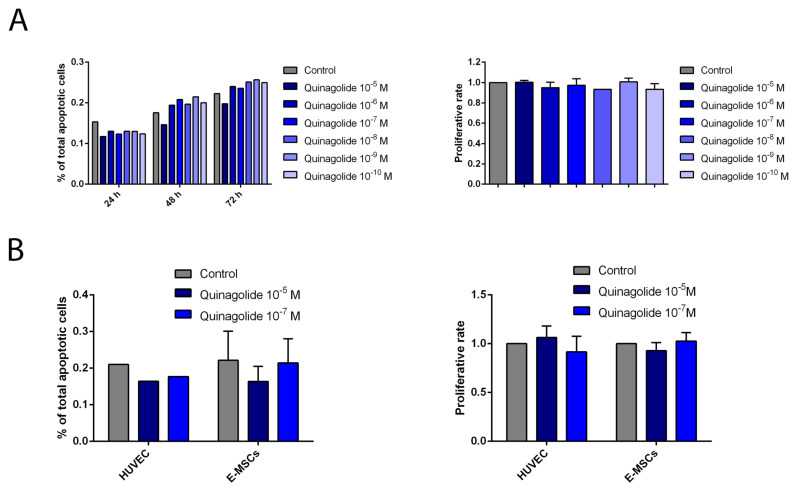
Quinagolide effect on E-MSC apoptosis and proliferation. (**A**) Quinagolide concentration-response curve on ectopic E-MSCs in both apoptosis (*n* = 1) and proliferation (*n* = 2) assays. (**B**) Effect of two selected quinagolide doses (10^−5^ and 10^−7^ M) on HUVECs (*n* = 1) and ectopic E-MSCs (*n* = 3) apoptosis and proliferation assays (*n* = 2). Data are represented as mean ± SD of the indicated number of experiments and normalized to untreated cells (Control).

**Figure 4 ijms-23-01775-f004:**
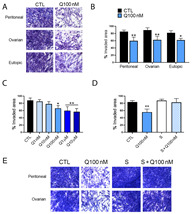
Quinagolide effect on E-MSC invasion. (**A**,**B**) Representative micrographs (**A**) and quantification (**B**) of quinagolide effect (100 nM) on eutopic and ectopic (both ovarian and peritoneal) E-MSC invasion (original magnification: X100). (**C**) Concentration response effect of quinagolide-treated ectopic E-MSC invasion. (**D**,**E**) Quantification (**D**) and representative micrographs (**E**) (original magnification: X100) of invasion assays performed on ectopic E-MSCs (both ovarian and peritoneal), treated with 100 nM quinagolide (Q100 nM), 5 µM spiperone (S) or a combination of quinagolide and spiperone (S + Q100 nM). All invasion data are represented as mean ± SD of at least three independent experiments, performed on different E-MSC lines, and normalized to untreated cells (CTL). One-way ANOVA was performed: * = *p* < 0.05 and ** = *p* < 0.001 vs. CTL.

**Figure 5 ijms-23-01775-f005:**
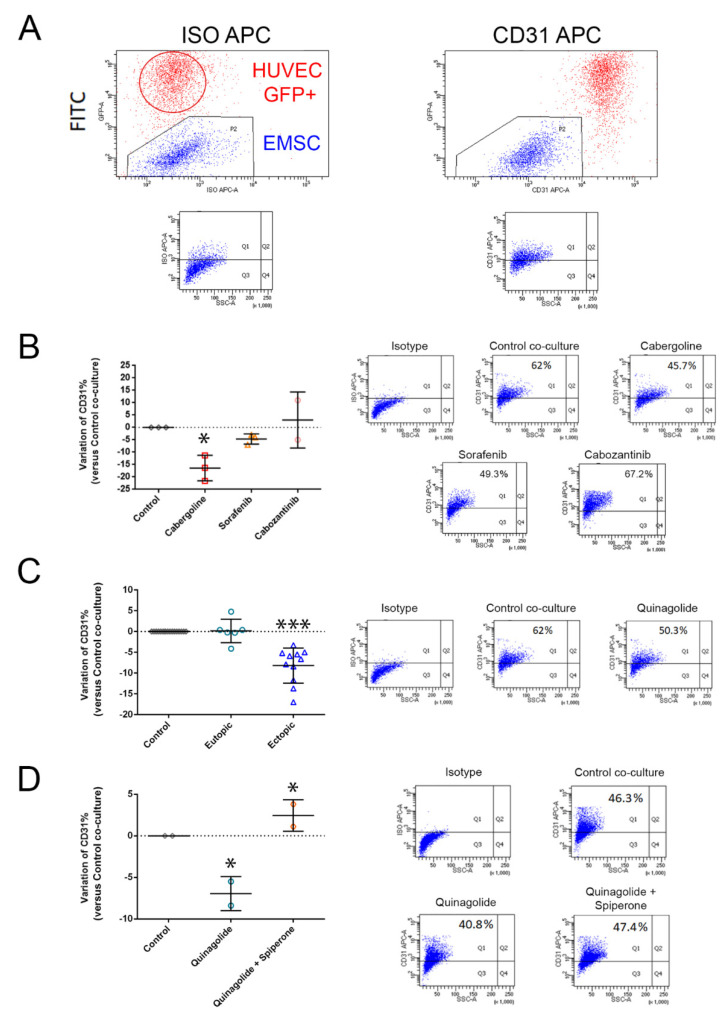
Quinagolide effect on E-MSC endothelial differentiation. (**A**) FACS gating strategy for the analysis of CD31^+^ E-MSCs after direct 48 h co-culture with HUVECs. E-MSCs are gated as GFP negative population, and CD31 expression is evaluated on the described gated E-MSCs after 48 h co-culture with HUVECs as percentage of CD31 APC positive events. (**B**–**D**) Representative flow cytometry micrographs and quantification, expressed as percentage of variation respect to control co-culture, of the effect of 24 h treatment of E-MSCs with 25 µM cabergoline, 1 µM sorafenib, 1 µM cabozantinib (**B**), 100 nM quinagolide (**C**) and the combination of 5 µM spiperone and 100 nM quinagolide (**D**) on E-MSC CD31 expression after 48 h of co-culture with HUVECs. Data are represented as mean ± SD of at least three independent experiments, performed on different ectopic E-MSC lines, and normalized to untreated co-culture. One-way ANOVA was performed: * = *p* < 0.05 and *** = *p* < 0.001 vs. control.

**Figure 6 ijms-23-01775-f006:**
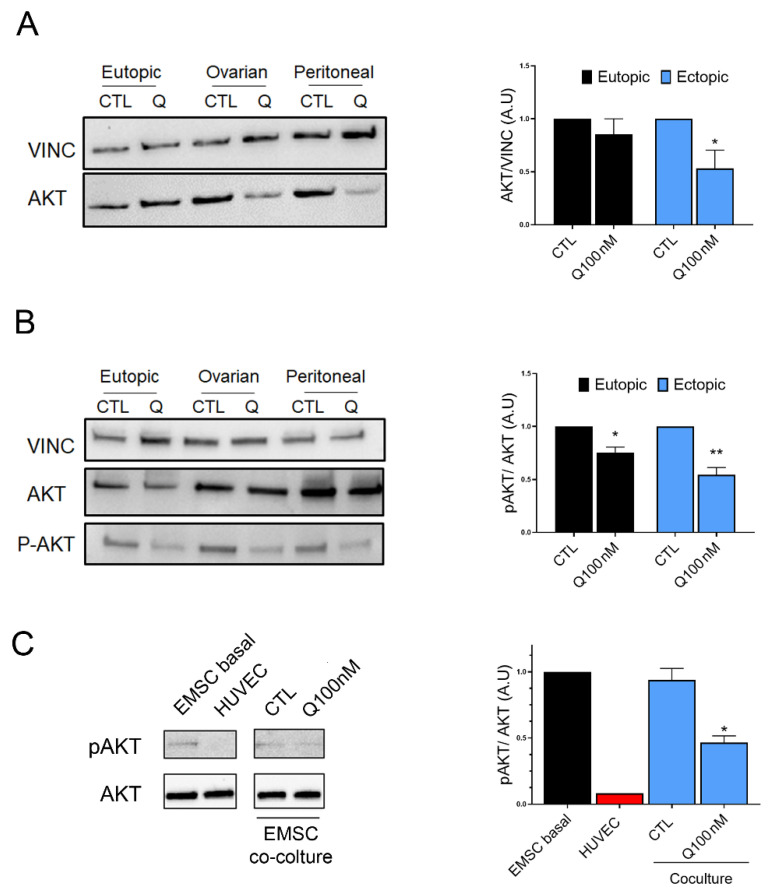
Quinagolide effect on AKT activation. (**A**): Representative western blot analysis and quantification of AKT levels in eutopic and ectopic E-MSCs treated for 48 h with 100 nM quinagolide (Q), compared to untreated cells (CTL). (**B**): Representative western blot analysis and quantification of P-AKT levels, normalized to AKT expression, in eutopic and ectopic E-MSCs treated for 48 h with 100 nM quinagolide (Q), with respect to untreated cells (CTL). (**C**): Representative western blot analysis and quantification of P-AKT levels in HUVECs, ectopic E-MSCs (E-MSC basal) and in E-MSCs after co-culture with HUVECs, treated or not for 24 h with 100 nM quinagolide (Q100 nM). Data are represented as mean ± SD of at least two independent experiments, performed on different E-MSC lines, and normalized to Vinculin and to untreated cells (CTL). One-way ANOVA was performed: * = *p* < 0.05 and ** = *p* < 0.001 vs. CTL.

**Table 1 ijms-23-01775-t001:** Demographic and clinical characteristics of patients enrolled in the study.

Patient #	Endometrial Samples	Age (Years)	Previous Pregnancies	Average Menstrual Cycle Length (Days)	Other Diseases
1	Eutopic	37	1	28–30	No
2	Eutopic	40	2	28	No
3	Eutopic	38	2	29	No
4	Ectopic (ovarian)	42	1	27	No
5	Ectopic (ovarian)	46	2	30	No
6	Ectopic (ovarian)	42	3	28	No
7	Ectopic (peritoneal and ovarian)	31	2	28	No
8	Ectopic (ovarian)	38	0	31	No
9	Ectopic (peritoneal and ovarian)	36	0	27	No
10	Ectopic (peritoneal)	40	1	31	No

Patient #: sequential number associated to each sample.

## Data Availability

All data will be made available after reasonable request to the corresponding author.

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
