# Peer review of "Quinagolide Treatment Reduces Invasive and Angiogenic Properties of Endometrial Mesenchymal Stromal Cells"

_ijms, 2022, doi:10.3390/ijms23031775_

Round 1

Reviewer 1 Report

The manuscript titled "Quinagolide treatment reduces invasive and angiogenic properties of Endometrial Mesenchymal Stromal Cells" is clearly described and is rich in content.
The Introduction section frames the general topic of the study and highlights the rationale for this research. In line 68 after the bibliographic citation 20 insert the square bracket and not the round one and in line 73 replace the square bracket with the round one.
The materials and methods are described clearly and completely with the appropriate bibliographic references.
Results are presented in logical sequence in the text and respect the objective of the work. 
Discussions are well described and supported by bibliographic studies.
Conclusions reflect the results obtained.

Author Response

The Introduction section frames the general topic of the study and highlights the rationale for this research. In line 68 after the bibliographic citation 20 insert the square bracket and not the round one and in line 73 replace the square bracket with the round one.

The materials and methods are described clearly and completely with the appropriate bibliographic references.
Results are presented in logical sequence in the text and respect the objective of the work. 
Discussions are well described and supported by bibliographic studies.
Conclusions reflect the results obtained.

Response:

We thank the reviewer for the suggestions, and for the positive comments.

We performed the suggested corrections on round and square brackets.

Reviewer 2 Report

 This is an experimental study where authors investigated the effect of quinagolide on E-MSCs isolated from eutopic endometrial tissue and ovarian and peritoneal endometriotic lesions and the mechanism involved.They concluded that dopamine receptor 2 activation led to downregulation of AKT and its phosphorylation and that inhibition of E-MSCs may increase the rationale for quinagolide in endometriosis treatment.

There are several points to be reconsidered before publication.

The abstract should obtain a more structured format, including all necessary sections.

In the introduction section, the gap in the literature should be highlighted together with the rationale of the current study.

In the materials and method section, clear eligibility criteria should be provided. Also, this section should be completely synchronized with the results, for the reader to follow the flow of the paper.

The results section should contain only results of the current study and not other references; these and the reporting could be moved to the discussion section.

Sample size calculation and limitations section are missing.

Author Response

This is an experimental study where authors investigated the effect of quinagolide on E-MSCs isolated from eutopic endometrial tissue and ovarian and peritoneal endometriotic lesions and the mechanism involved. They concluded that dopamine receptor 2 activation led to downregulation of AKT and its phosphorylation and that inhibition of E-MSCs may increase the rationale for quinagolide in endometriosis treatment. There are several points to be reconsidered before publication.

Response. We thank the reviewer for the suggestions, that we think improved our manuscript.

1. The abstract should obtain a more structured format, including all necessary sections.

Response. We agree with the reviewer and substantially changed the abstract, adding information in a more structured format. However, the continuous abstract structure of the journal style does not allow different sections.

2. In the introduction section, the gap in the literature should be highlighted together with the rationale of the current study.

Response. We thank the reviewer for the suggestion and we highlighted the gap in the literature together with the rationale of the current study. See Introduction, lines 84-90.

3. In the materials and method section, clear eligibility criteria should be provided. Also, this section should be completely synchronized with the results, for the reader to follow the flow of the paper.

Response. We added the eligibility criteria for patient inclusion in the study. See lines 338-343. The Material and Method section was moved at the end of the manuscript to comply with the journal style. Moreover, the sub-chapters have been reordered to better follow the results, as suggested.

4. The results section should contain only results of the current study and not other references; these and the reporting could be moved to the discussion section.

Response. We agree with the reviewer and we removed the references from the results together with the related comments where appropriate. Only two references were left in the Results as they were needed for the readers to better understand the rationale for the experiments performed. See line 148.

5. Sample size calculation and limitations section are missing.

Response. We added the sample size and the limitation of the study, as requested. See lines 452-455 and 324-328.

Reviewer 3 Report

Well written, interesting topic 

Scientific approach for incurable endometriosis disease and mechanism

Minor point:

Abstract Line 4:in endometrial?

Table 1: Tissue preparation Endometrial samplesàHow much in sample tissue? In cancer tissue sample and pathologic marker study percentage of cancer fraction in sample is important. I’m not sure of ovarian, peritoneal tissue ES proportion and accurate sample quality. How about using patient’s blood sample or menstrual tissue?

2.5: Drugs and Reagents: Cabergoline, sorafenib and cabozantinib are mentioned. How about using other potent VEGF related target treatment like Bevacizumab and Cediranib .and additional explanation about their relative potency regarding VEGF affinity is needed  

Author Response

Abstract Line 4:in endometrial?

Response. We thank the reviewer for pointing out this mistake. We corrected endometrial into " in models of endometriosis".

1-Table 1: Tissue preparation Endometrial samples How much in sample tissue? In cancer tissue sample and pathologic marker study percentage of cancer fraction in sample is important. I’m not sure of ovarian, peritoneal tissue ES proportion and accurate sample quality. How about using patient’s blood sample or menstrual tissue?

Response.

We now indicate that the size of the endometriotic tissue used to generate EMSCs was around 0.5 cm3. (see line 350). We agree that the percentage of cancer fraction in sample is important when using cancer tissue samples (where central necrosis can be present) and in pathologic marker studies. However, we considered in our case this aspect to be not so relevant as the samples were instrumental to generate cell lines of E-MSCs, and the experiments were normalized on the number of cells. Indeed, the tissue proportion was low but sufficient to generate cell lines. Moreover, the sample quality was high, as it is a highly vascularized tissue, easily recognizable when present on peritoneum or ovaries.

The possibility of using menstrual tissue is of value for patients with endometriosis, however, this would not allow to obtain the ectopic peritoneal and ovarian endometriotic tissue. Of interest, we could show the expression of DRD2 receptor on these cells, supporting the rationale for the quinagolide therapy.

2: Drugs and Reagents: Cabergoline, sorafenib and cabozantinib are mentioned. How about using other potent VEGF related target treatments like Bevacizumab and Cediranib, and additional explanation about their relative potency regarding VEGF affinity is needed  

Response.

We thank the reviewer for the insights. Sorafenib and cabozantinib, anti-angiogenic tyrosine kinase inhibitors, were used to assess the potential contribution of VEGFR2 inhibition on the effects of quinagolide, a DRD2 agonist also shown to affect VEGFR2 signaling. As Sorafenib and Cabozantinib had no effect on endothelial differentiation, we could speculate on the relevance of DRD2 activation in the endothelial differentiation model. This has now been better discussed (lines 301-304).

Reviewer 4 Report

The authors in this manuscript attempted to elucidate the effect of quinagolide, a dopamine receptor 2 agonist, on endometrial mesenchymal stromal cells (E-MSCs) isolated from eutopic endometrial tissue as well as ovarian and peritoneal endometriotic lesions. They found that quinagolide inhibited the invasive properties of E-MSCs, and limited their endothelial differentiation in an endothelial co-culture model of angiogenesis. Moreover, dopamine receptor 2 activation led to downregulation of AKT and its phosphorylation. These results indicate the therapeutic potential of quinagolide in endometriosis treatment. The topic is clinically significant and interesting. The finding in this manuscript is convincing and the manuscript is well written. I have only a small question regarding the supplementary materials. Although supplementary figures 1A and 1B have been mentioned in the section of results, the figure legends or titles were not listed in the paragraph “Supplementary material” (Line 433-434).

Author Response

The authors in this manuscript attempted to elucidate the effect of quinagolide, a dopamine receptor 2 agonist, on endometrial mesenchymal stromal cells (E-MSCs) isolated from eutopic endometrial tissue as well as ovarian and peritoneal endometriotic lesions. They found that quinagolide inhibited the invasive properties of E-MSCs, and limited their endothelial differentiation in an endothelial co-culture model of angiogenesis. Moreover, dopamine receptor 2 activation led to downregulation of AKT and its phosphorylation. These results indicate the therapeutic potential of quinagolide in endometriosis treatment. The topic is clinically significant and interesting. The finding in this manuscript is convincing and the manuscript is well written. I have only a small question regarding the supplementary materials. Although supplementary figures 1A and 1B have been mentioned in the section of results, the figure legends or titles were not listed in the paragraph “Supplementary material” (Line 433-434).

Response. We thank the reviewer for the suggestions, and for the positive comments. We now added the title in the paragraph “Supplementary material” (Line 468).

Round 2

Reviewer 2 Report

I think that all suggestions have been addressed.